# Extraction of clinical data on major pulmonary diseases from unstructured radiologic reports using a large language model

Hyung Jun Park[1]☯, Jin-Young Huh[2]☯, Ganghee Chae[3]*, Myeong Geun Choi[4]*

1 Department of Internal Medicine, Division of Pulmonary and Critical Care Medicine, Shihwa Medical Center, Siheung, Korea, 2 Department of Internal Medicine, Division of Pulmonary, Allergy and Critical Care Medicine, Chung-Ang University Gwangmyeong Hospital, Gwangmyeong, Korea, 3 Department of Internal Medicine, Division of Pulmonary and Critical Care Medicine, Ulsan University Hospital, University of Ulsan College of Medicine, Ulsan, Korea, 4 Department of Internal Medicine, Division of Pulmonary and Critical Care Medicine, Mokdong Hospital, College of Medicine, Ewha Womans University, Seoul, Korea

☯ These authors contributed equally to this work.
* cmkcmk1006@gmail.com (MGC); ganghee@uuh.ulsan.kr (GC)

**Data Availability Statement:** All relevant data are within the manuscript and its Supporting Information files.

## Abstract

Despite significant strides in big data technology, extracting information from unstructured clinical data remains a formidable challenge. This study investigated the utility of large language models (LLMs) for extracting clinical data from unstructured radiological reports without additional training. In this retrospective study, 1800 radiologic reports, 600 from each of the three university hospitals, were collected, with seven pulmonary outcomes defined. Three pulmonology-trained specialists discerned the presence or absence of diseases. Data extraction from the reports was executed using Google Gemini Pro 1.0, OpenAI's GPT-3.5, and GPT-4. The gold standard was predicated on agreement between at least two pulmonologists. This study evaluated the performance of the three LLMs in diagnosing seven pulmonary diseases (active tuberculosis, emphysema, interstitial lung disease, lung cancer, pleural effusion, pneumonia, and pulmonary edema) utilizing chest radiography and computed tomography scans. All models exhibited high accuracy (0.85–1.00) for most conditions. GPT-4 consistently outperformed its counterparts, demonstrating a sensitivity of 0.71–1.00; specificity of 0.89–1.00; and accuracy of 0.89 and 0.99 across both modalities, thus underscoring its superior capability in interpreting radiological reports. Notably, the accuracy of pleural effusion and emphysema on chest radiographs and pulmonary edema on chest computed tomography scans reached 0.99. The proficiency of LLMs, particularly GPT-4, in accurately classifying unstructured radiological data hints at their potential as alternatives to the traditional manual chart reviews conducted by clinicians.

## Introduction

Recent advancements in big data technology have prompted many centers to develop their own datasets for specific diseases or establish clinical data warehouses, facilitating the extraction of patients clinical data [1]. The advancements in large language models (LLMs) have transcended patient classification and data structuring, extending into diverse medical fields,

**Funding:** The author(s) received no specific funding for this work.

**Competing interests:** The authors have declared that no competing interests exist.

including education, medical counselling, and clinical practice. These AI models have demonstrated their potential in enhancing medical education by aiding students in preparing for the USMLE exam [2] and solving NEJM quizzes [3]. Furthermore, ChatGPT has been used to reduce the communication gap between patients and healthcare professionals, as well as facilitating patient triage before using medical facilities [4]. In clinical settings, LLMs help healthcare professionals understand medical reports [5,6], assist in diagnosing patient conditions [7], and further contribute to clinical decision-making processes [8,9].

Despite the expansion of data size, a notable gap exists in methods for efficiently locating major patient concerns, such as treatment progress and patient-reported symptoms. This gap has resulted in a reliance on indirect methodologies, such as International Classification of Diseases codes or medication usage data, to investigate disease manifestations. In cases where data analysis is not feasible, it becomes essential to manually review the medical records of all patients within the cohort. To address this challenge, the proposed methods leverage unstructured medical records and apply artificial intelligence to extract crucial research variables, complications, and disease groups [10–13]. Our team previously conducted studies attempting to derive clinical data from X-ray and positron emission tomography scans [14,15]. However, these efforts were hindered by the need for new data each time a new model was trained, and the annotation of new data required additional loading, making it challenging. Consequently, the model failed to effectively replace clinicians' reviews of medical reports.

The advanced models ChatGPT and GPT-4 have been employed across various fields, and their Application Programming Interfaces are now publicly accessible, enabling the use of natural language processing models at a low cost ($0.002 per 1K token) [16]. In non-medical fields, these models have shown higher accuracy and inter-observer agreement than human curation [17]. Efforts are also being made to use GPT-4 for data labelling in the medical field, including pulmonary diseases [18,19]. However, the models have only been validated in situations with detailed radiology reports similar to those found in Anglophone countries without evaluating the characteristics of data written by Koreans, which tend to be shorter compared to those from predominantly English-speaking regions. Therefore, it is crucial to verify whether the performance of these models is maintained in reports written in countries where English is not the native language or in reports where English and Korean are mixed; however, such verification is currently lacking.

Therefore, this study aimed to investigate the usefulness and accuracy of LLMs for extracting clinical data from unstructured radiological reports without additional training.

## Materials and methods

### Data collection

Radiological reports for chest computed tomography (CT) and chest radiography were gathered from 300 inpatients in the pulmonology departments of Ewha Womans University Mokdong Hospital (EUMC), Ulsan University Hospital (UUH), and Chung-Ang University Gwangmyeong Hospital (CAUGH) between March 2022 and February 2023 (Fig 1). The data were accessed on January 12, 2024, at EUMC, on January 17, 2024, at UUH, and on February 11, 2024, at CAUGH. Reports were included consecutively until the total number of X-ray and CT scans reached 300, irrespective of specific disease occurrences.

This study received approval from the Institutional Review Boards of Ewha Womans University Mokdong Hospital (approval number: 2023-07-002), Ulsan University Hospital (approval number: 2023-12-011), and Chung-Ang University Gwangmyeong Hospital (approval number: 2312-127-143) and was conducted in accordance with the ethical standards

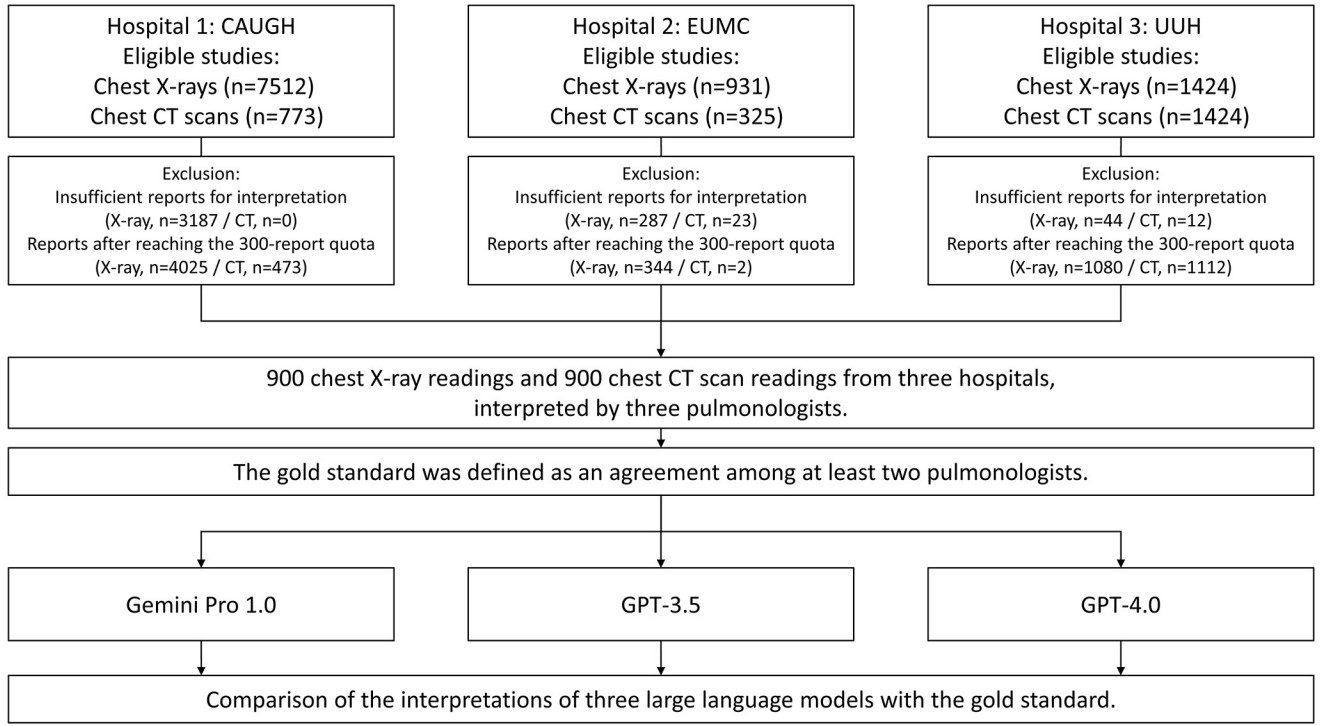

**Fig 1. Flowchart of the study.**

outlined in the Declaration of Helsinki. The necessity for written informed consent was waived due to the retrospective nature of the study.

## Inclusion and exclusion criteria

During the study period, the first radiographic or CT scans of inpatients admitted to the pulmonology department were selected. Reports lacking sufficient or suitable content for interpretation were omitted to ensure the quality and relevance of our dataset, as such reports could lead to an overestimation of model performance by representing overly simplistic tasks that do not reflect the complexity of real-world clinical scenarios. These comprised reports including "no active lung lesions", "hypoinflated lung", "no significant interval changes", "no abnormalities", "negative chest findings", "within normal limits", "simple postoperative status", and "unremarkable findings".

## Outcome definition and labelling

The following outcomes were defined: pneumonia, interstitial lung disease (ILD), active pulmonary tuberculosis (TB), pulmonary edema, pleural effusion, lung cancer, and emphysema. On CT scans, interstitial lung abnormalities were identified as ILD, and lung abscesses were classified as pneumonia. The complete resolution and improvement were recorded as absent, whereas interval improvement and decreases were recorded as present. On chest radiographs, pulmonary congestion was assessed for edema, and interstitial pneumonia was categorized as ILD. Three pulmonology specialists, GC, MGC, and J-YH, each with over a decade of experience as practicing physicians including more than 5 years each of clinical experience in pulmonology, independently reviewed the reports and labeled the presence or absence of seven

diseases. All three hold assistant professorships at university hospitals with clinical and research expertise. GC's clinical and research areas include tuberculosis (mycobacterial disease), lung cancer, and interventional pulmonology; MGC specializes in lung cancer and interventional pulmonology; and J-YH's areas of expertise are lung cancer and interstitial lung disease. The gold standard was established as an agreement among at least two labelers.

### Model selection

The latest models available were used in this study, including Google's Gemini Pro 1.0 and OpenAI's GPT-3.5 and GPT-4, specifically the gpt-3.5-turbo-1106 and gpt-4-1106-preview versions. Data were extracted using these models, with the prompts listed in S1 Table.

### Statistical analysis

The reports were analyzed using Gemini Pro 1.0, GPT-3.5, and GPT-4, maintaining a consistent temperature of 0.3 across all models. The performance of these models in accurately interpreting reports, compared to the gold standard, was assessed in terms of accuracy, sensitivity, and specificity. For each of the seven diseases and each model (Gemini Pro 1.0, GPT-3.5, and GPT-4), we calculated sensitivity, specificity, and accuracy by comparing the model's predictions against the gold standard established by agreement between pulmonologists. These metrics were computed separately for each disease category, allowing for a detailed assessment of model performance across different clinical entities. This approach enabled us to evaluate how well each model identified both the presence and absence of specific diseases in the radiologic reports, providing a comprehensive view of their diagnostic capabilities.

Additionally, the accuracy of each label was evaluated against the gold standard using these metrics, stratified by disease and hospital, to determine the accuracy and appropriateness of human annotation. Interobserver agreement among the labelers was assessed by calculating the Fleiss kappa value for each disease. The output of each model was evaluated for adherence to the specified JSON format, which included the correct structure and presence of all seven disease fields. Errors in JSON formatting were recorded and analyzed to assess the models' ability to consistently produce structured output.

## Results

### Incidence of pulmonary conditions in this study

Table 1 presents the distribution of various pulmonary conditions found in the radiography and CT reports from three hospitals (CAUGH, UUH, and EUMC). Pneumonia emerged as

**Table 1. Incidence rates of pulmonary conditions by hospital and imaging modality.**

| Hospital | X-ray | | | CT | | |
|---|---|---|---|---|---|---|
| | **CAUGH** | **UUH** | **EUMC** | **CAUGH** | **UUH** | **EUMC** |
| **Pneumonia** | 48.00 | 57.67 | 61.67 | 39.67 | 53.33 | 41.00 |
| **ILD** | 7.33 | 6.00 | 5.33 | 13.00 | 14.00 | 6.33 |
| **Active TB** | 2.33 | 1.67 | 2.00 | 7.00 | 5.33 | 5.33 |
| **Pulmonary edema** | 6.00 | 7.67 | 14.67 | 1.67 | 8.33 | 7.67 |
| **Pleural effusion** | 23.33 | 23.33 | 40.00 | 27.00 | 37.00 | 34.67 |
| **Lung cancer** | 13.33 | 10.67 | 3.67 | 42.00 | 21.33 | 20.00 |
| **Emphysema** | 5.67 | 23.00 | 5.67 | 27.00 | 21.33 | 20.33 |

CT, computed tomography; CAUGH, Chung-Ang University Gwangmyeong Hospital; UUH, Ulsan University Hospital; EUMC, Ewha Womans University Mokdong Hospital; ILD, interstitial lung disesase; TB, tuberculosis.

the most prevalent condition (39.67–61.67%) across hospitals and imaging modalities. Following closely, pleural effusion ranked as the second most prevalent condition (23.33–40.00%). Notably, the incidence of lung cancer exhibited a higher frequency in CT reports (20.00–42.00%), compared with X-ray reports (3.67–13.33%). Moreover, the prevalence of patients with ILD, active TB, pulmonary edema, and emphysema variations across hospitals and imaging modalities.

## Performance of AI models in radiologic report analysis

Table 2 presents an overview of how the three AI models (Gemini Pro 1.0, GPT-3.5, and GPT-4) performed in analyzing X-ray and CT reports. Across various scenarios, all models exhibited remarkable accuracy, sensitivity, and specificity. Notably, GPT-4 emerged as a top performer, achieving a sensitivity range of 0.71 to 1.00 in chest X-rays and 0.90 to 1.00 in CT scans, alongside a specificity of 0.89–1.00 and an accuracy of 0.89–0.99 across both modalities. This underscores its superior capability in interpreting radiological reports. Gemini Pro 1.0 and GPT-3.5 also showed strong performance, with almost all metrics surpassing 0.80.

However, Fig 2 shows that the accuracy of the models varied based on the type of disease and the hospital for both CT and X-ray reports. Specifically, for ILD in CT reports, Gemini Pro 1.0 and GPT-4 exhibited higher sensitivity in detecting the condition but slightly lower specificity and accuracy compared to GPT-3.5. Conversely, regarding pulmonary edema in CT reports, GPT-3.5 showed notably inferior performance across all hospitals compared to the other models.

Regarding X-ray reports (as shown in Fig 3), the models' performance varied across hospitals and diseases. At UUMC hospital, all models exhibited decreased performance in detecting pneumonia, ILD, and pulmonary edema compared to other hospitals. Particularly, for active TB, GPT-3.5 demonstrated significantly lower sensitivity than the other models at UUMC hospital. It is also crucial to note that all models had a sensitivity of 0 for detecting pulmonary edema in X-ray reports from UUMC Hospital.

**Table 2. Performance metrics of large language models in diagnosing pulmonary conditions from radiologic reports.**

| Report | Condition | Gemini Pro 1.0 | | | GPT 3.5 | | | GPT 4.0 | | |
|---|---|---|---|---|---|---|---|---|---|---|
| | | Sensitivity | Specificity | Accuracy | Sensitivity | Specificity | Accuracy | Sensitivity | Specificity | Accuracy |
| X-ray | Pneumonia | 0.85 | 0.96 | 0.90 | 0.75 | 0.98 | 0.85 | 0.87 | 0.94 | 0.90 |
| | Interstitial lung disease | 0.93 | 0.90 | 0.91 | 0.96 | 0.89 | 0.90 | 0.96 | 0.89 | 0.89 |
| | Active tuberculosis | 0.83 | 0.97 | 0.96 | 0.78 | 0.96 | 0.95 | 0.94 | 0.95 | 0.95 |
| | Pulmonary edema | 0.68 | 1.00 | 0.97 | 0.64 | 1.00 | 0.96 | 0.71 | 1.00 | 0.97 |
| | Pleural effusion | 0.98 | 0.97 | 0.97 | 0.96 | 0.97 | 0.97 | 1.00 | 0.99 | 0.99 |
| | Lung cancer | 0.81 | 0.96 | 0.95 | 0.82 | 0.97 | 0.96 | 0.95 | 0.96 | 0.96 |
| | Emphysema | 0.98 | 1.00 | 1.00 | 1.00 | 0.99 | 1.00 | 1.00 | 0.99 | 0.99 |
| CT | Pneumonia | 0.91 | 0.88 | 0.89 | 0.84 | 0.93 | 0.89 | 0.90 | 0.91 | 0.91 |
| | Interstitial lung disease | 0.90 | 0.94 | 0.94 | 0.87 | 0.94 | 0.93 | 0.94 | 0.88 | 0.89 |
| | Active tuberculosis | 0.92 | 0.94 | 0.94 | 0.74 | 0.93 | 0.92 | 0.96 | 0.95 | 0.95 |
| | Pulmonary edema | 0.92 | 0.99 | 0.99 | 0.89 | 1.00 | 0.99 | 0.94 | 1.00 | 0.99 |
| | Pleural effusion | 0.97 | 0.94 | 0.95 | 0.97 | 0.93 | 0.94 | 0.98 | 0.96 | 0.96 |
| | Lung cancer | 0.94 | 0.89 | 0.90 | 0.88 | 0.91 | 0.90 | 0.97 | 0.88 | 0.90 |
| | Emphysema | 1.00 | 0.97 | 0.98 | 0.99 | 0.97 | 0.97 | 1.00 | 0.96 | 0.97 |

CT, computed tomography.

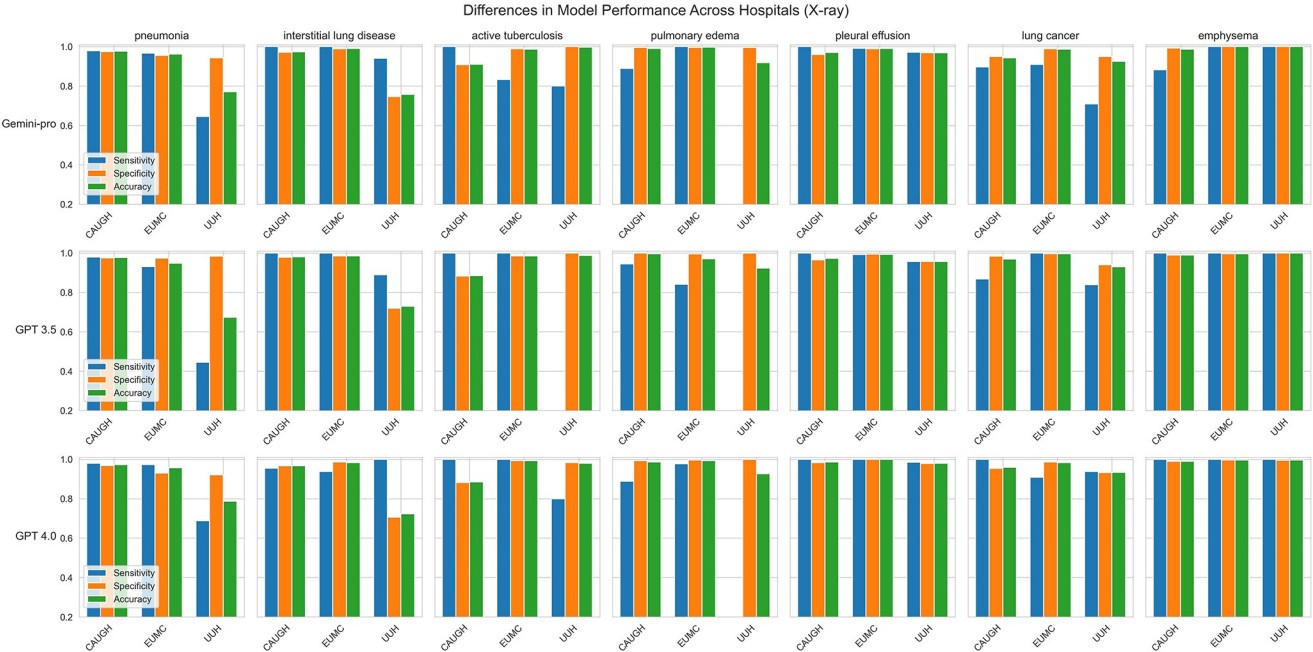

**Fig 2. Comparative performance of the models in diagnosing pulmonary conditions using chest X-ray reports across hospitals.**

## Interobserver agreement among pulmonologists

The interobserver agreement among the human labelers varied across diseases and hospitals, both in X-ray and CT reports. Fleiss's kappa values ranged from 0.384 to 0.941 for chest X-rays and from 0.556 to 0.870 for CT (Figs 4 and 5 and S2 Table). This suggests that experienced

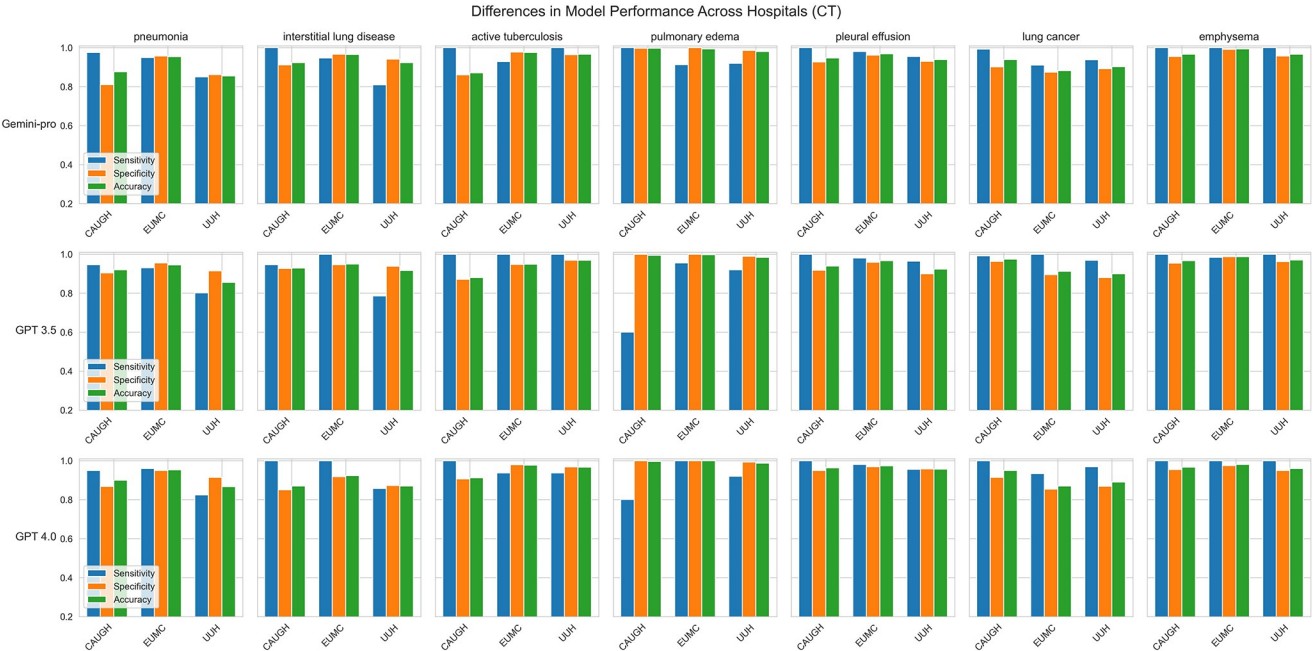

**Fig 3. Comparative performance of the models in diagnosing pulmonary conditions using chest CT reports across hospitals.**

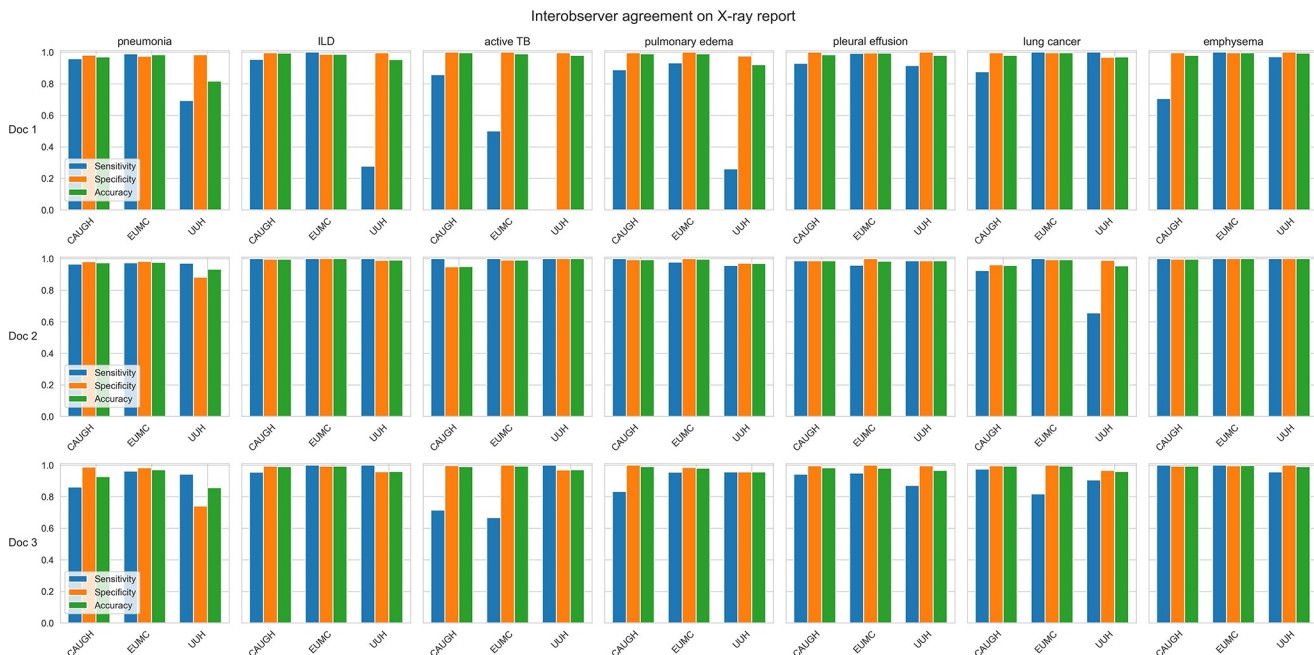

**Fig 4. Interobserver agreement on diagnosing pulmonary conditions using chest X-ray reports across hospitals.**

physicians may differ in their assessments of certain conditions. For CT reports, agreement was notably low for ILD at CAUGH, implying differences in the interpretation of ILD findings among labelers at this hospital. Moreover, agreement for active TB was generally lower across all hospitals, indicating potential disparities in how the labelers interpreted and extracted

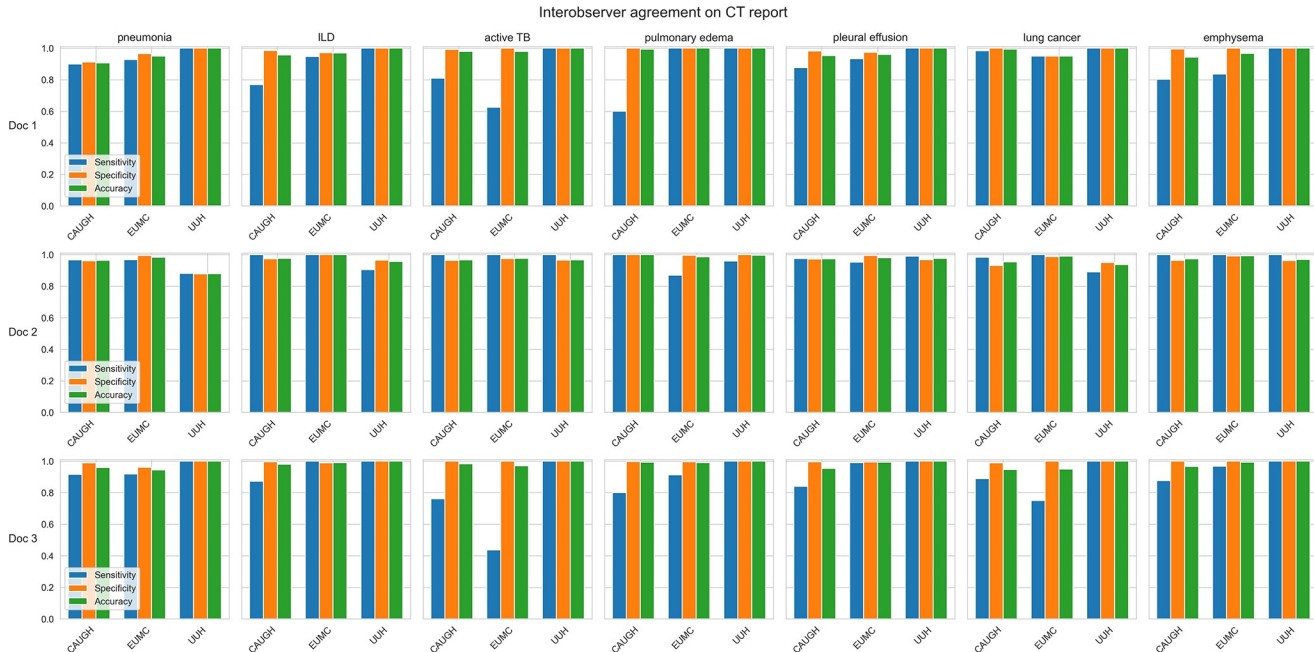

**Fig 5. Interobserver agreement on diagnosing pulmonary conditions using chest CT reports across hospitals.**

information related to active TB from CT reports. Regarding X-ray reports (Fig 5), interobserver agreement varied significantly for ILD, active TB, and pulmonary edema across hospitals and labelers, highlighting potential differences in how labelers analyzed and comprehended X-ray reports under these circumstances.

### Error rates in JSON format generation by AI models

Table 3 presents the error rates for the models used in generating JSON format accurately. GPT-4 had no errors on the first attempt, whereas Gemini Pro 1.0 and GPT-3.5 had error rates of 6.8% and 4.7%, respectively. Subsequent evaluations focused solely on erroneous cases. By the third attempt, Gemini Pro 1.0 reduced its error rates to 1.1%, and GPT-3.5 decreased to 3.2%, considering only the initially erroneous cases.

## Discussion

This study introduces an innovative approach utilizing LLMs, including GPT-4, to extract clinical data from unstructured radiologic reports and categorize the presence of major respiratory diseases, thus organizing the data effectively. In comparison to the gold standard results manually categorized by three pulmonologists, GPT-4 exhibited accuracy of 0.89–0.99, indicating exceptional consistency across most respiratory conditions. Notably, the accuracy of pleural effusion and emphysema on chest radiographs and pulmonary edema on chest CT scans reached 0.99.

Since the development of GPT-4, two studies have utilized this model to analyze unstructured information in respiratory diseases. Lisa et al. used GPT-4 to perform a binary classification of key findings from unstructured interpretations of chest CTs and magnetic resonance imaging (MRIs) [19]. They successfully converted these findings into structured data, achieving high F1 scores (a measure combining recall and precision) for venous catheters, tracheal tubes, and thoracic drains. Matthias et al. classified the oncological phenotypes of patients based on free-text descriptions in lung cancer CT scans using GPT-4 [18]. This study involved 424 patients with lung cancer and extracted data on tumor size, location, metastatic lesions, and disease assessment using LLMs and comparing these results with manual classifications by four radiologists. GPT-4 demonstrated excellent performance, with accuracies between 90.0 and 100.0%. However, these studies were limited to structuring simple findings described in free text rather than disease impressions. Similarly, our study utilized the latest LLM to achieve results comparable to manual human categorization, bolstering the utility of LLMs and extending their use not only to radiological findings but also to classify the presence of major respiratory diseases.

Despite the overall excellent performance, the lowest accuracy was observed in classifying pneumonia and ILD from X-rays and ILD and lung cancer from CT scans. Diagnosing these diseases requires clinical correlation, and they can exhibit various radiological findings [20–22]. Moreover, clinical experience and other factors can influence the diagnosis, suggesting that the accuracy of LLM models may be lower when classifying these diseases.

**Table 3. Error rates of the model for generating appropriate JSON format.**

| Model | First attempt | Second attempt | Third attempt |
|---|---|---|---|
| Gemini Pro 1.0 | 62/900 (6.8%) | 12/900 (1.3%) | 10/900 (1.1%) |
| GPT 3.5 | 43/900 (4.7%) | 33/900 (3.6%) | 29/900 (3.2%) |
| GPT-4 | 0/900 (0%) | | |

For ILD, the specificity was relatively low (0.89 in X-ray and 0.88 in chest CT). ILD can be classified into many different types depending on the cause, clinical presentation, and histological findings. The imaging spectrum is highly variable, including honeycombing, reticular opacity, traction bronchiectasis, fibrosis, ground-glass opacity, consolidation, nodules, and cystic lesions [20]. However, similar imaging findings, such as fibrosis, ground-glass opacity, and consolidation, can also appear in cases of infection, drug reactions, or radiation therapy. Thus, comprehensive assessment of the clinical context is essential when confirming ILD. Similarly, pneumonia cannot be diagnosed based solely on imaging findings and must be accompanied by clinical signs and symptoms [21]. This highlights that LLMs are not yet fully capable of considering clinical correlations, which may explain the reduced accuracy of the LLM models in these conditions.

We analyzed cases where GPT-4 categorized them as ILD, although pulmonologists had diagnosed them otherwise. The most common case, accounting for 53.7% of these instances, occurred when GPT-4 classified any findings of fibrosis on chest CT, regardless of the cause (e.g., radiotherapy, COVID, tuberculosis), as ILD, whereas pulmonologists did not. Additionally, due to the challenge of diagnosing ILD solely on one finding, pulmonologists did not classify cases of interlobular septal thickening, which could also indicate pulmonary edema or lymphangitic metastasis, as ILD. Conversely, GPT-4 classified them as ILD (22.1%). Furthermore, there was low sensitivity in classifying pulmonary edema from chest radiographs. While pulmonologists identified all instances of interstitial opacity with or without pleural effusion, such as pulmonary edema, GPT-4 did not. Notably, a previous study reviewed 50 chest radiology reports, where four radiologists formed impressions from the radiology reports, compared to impressions generated by GPT-4 [23]. The evaluation revealed that impressions created by radiologists scored significantly higher in coherence, factual consistency, comprehensiveness, and medical harmfulness than those created by GPT-4, highlighting the limitations of AI in tasks that require clinical experience. Similarly, our research suggests that AI has limitations in classifying diseases that require comprehensive consideration during diagnosis.

On the other hand, in some cases, the LLM appeared to correlate diagnostic criteria with the context of the radiologic report and avoided making judgments on items with clinical uncertainty, that is, items in the report lacking definitive evidence for diagnosis. For example, upon reviewing the decisions made by GPT-4, it did not classify "developing and increasing in size of heterogeneously enhancing nodules in the right lower lobe" as lung cancer on a CT scan of a patient with breast cancer. In a case where the report mentioned the possibility of chronic obstructive pulmonary disease—a condition where emphysema may appear in radiologic findings but is not essential for diagnosis [24]—in a patient with chronic bronchitis, GPT-4 did not classify the finding as emphysema. Furthermore, the LLM interpreted medical abbreviations within the context of the entire radiologic report, making inferences regardless of whether abbreviations or full terms were used. For instance, GPT-4 correctly recognized all instances of "TB" as "tuberculosis" in the reports. We did not perform any specific preprocessing to standardize acronym usage, as the models demonstrated the ability to handle this variation naturally. These capabilities are part of the models' strength in processing natural language as it appears in real-world medical reports, contributing to its effectiveness in our study.

In addition, our study found that interobserver agreement was lowest for active pulmonary TB, likely due to the difficulty in determining disease activity solely from radiological findings [25]. This complexity is further compounded by the pivotal role of microbiological test results in confirming diagnoses, leading to differing interpretations among observers. For instance, discrepancies arose when some observers identified the presence of a cavitary nodule in the upper lobe on radiographs as indicative of active TB, while others did not

consider this as conclusive evidence. Utilizing AI for classifying diseases that require personal clinical experience and subjective judgment could provide more consistent results compared to human evaluators. However, thorough validation is crucial, particularly in cases involving complex medical decision-making. Furthermore, differences in observer agreement were observed across various centers, with classifications by pulmonologists at these centers often closely aligning with the gold standard. This implies that observers' familiarity with and adaptation to the radiologic reading practices of their respective centers may have influenced these findings.

Although all AI models demonstrated strong agreement with human annotators in terms of performance, notable discrepancies were observed in their stability when generating JSON formats. GPT-4 showcased exceptional stability, flawlessly producing error-free JSON formats on its first attempt. Error-free JSON formats refer to outputs that strictly adhered to the specified structure, including all seven disease fields with appropriate "yes" or "no" values, without any syntactical errors or missing information. Conversely, Gemini Pro 1.0 and GPT-3.5 exhibited error rates of 6.8% and 4.7%, respectively. These errors ranged from improper JSON formatting or providing answers solely for specific diseases like pneumonia, resulting in parsing challenges. While error rates decreased with subsequent attempts, the initial stability of GPT-4 highlights its superiority in both performance and reliability, making it the optimal model for clinical data extraction and structuring despite its higher cost. Our findings align with previous studies indicating superior performance with GPT-4 compared to earlier models [26,27]. Additionally, we present a comparison of error rates in JSON format generation across each model.

However, this study has some limitations. Firstly, it relied on labeler classification as the gold standard, which may introduce bias as human decisions can diverge from actual patient diagnoses. Nonetheless, the study aimed to ascertain whether AI could effectively replace human classification and to what extent it aligns with human decisions, enabling us to derive meaningful insights. A second limitation is that the pulmonologists made their classifications solely on free text, without direct access to the images. Although this approach was tailored to current AI technology, future advancements may necessitate investigations into whether AI can directly classify patients through image analysis. Moreover, our research utilized radiologic reports written by Korean radiologists, which may limit its generalizability to regions with different reporting styles and medical practices [28]. Future studies could benefit from validating these findings using datasets from other countries and linguistic backgrounds to assess the models' performance across diverse contexts. Lastly, the models' performance in this study may have been slightly compromised due to the lack of recent advancements in prompt design techniques. The potential impact of prompt design approaches such as chain of thought, reflection, and few-shot in-context learning, which have shown promise in improving model performance, was not evaluated in this study. Consequently, further research is required to validate these prompt design techniques and enhance the accuracy and reliability of AI-assisted classification in medical contexts.

In conclusion, this study demonstrated the capability of LLMs, particularly GPT-4, to efficiently extract clinical data from unstructured radiologic reports without the need for additional training in respiratory research. The capability is accompanied by high accuracy, sensitivity, and specificity. These results suggest the potential of automating data extraction processes to improve clinical data repositories and facilitate large-scale medical studies. The proficiency of LLMs, particularly GPT-4, in accurately classifying unstructured radiological data hints at their potential as alternatives to traditional manual chart reviews conducted by clinicians. However, the variability in performance observed across diseases and healthcare facilities, as well as disparities in human labelling, highlights the need for further research,

establishment of standardized guidelines, and comprehensive training to ensure the robustness of models and consistent interpretation of data.

## Supporting information

**S1 Table. System and user prompts for extracting pulmonary outcomes from radiologic reports.**
(DOCX)

**S2 Table. Fleiss's Kappa value for interobserver agreement among pulmonologists.**
(DOCX)

**S1 File.**
(XLSX)

## Author Contributions

**Conceptualization:** Hyung Jun Park.

**Data curation:** Hyung Jun Park, Jin-Young Huh, Ganghee Chae, Myeong Geun Choi.

**Formal analysis:** Hyung Jun Park, Jin-Young Huh.

**Investigation:** Hyung Jun Park, Jin-Young Huh, Ganghee Chae, Myeong Geun Choi.

**Methodology:** Hyung Jun Park, Jin-Young Huh, Ganghee Chae, Myeong Geun Choi.

**Writing – original draft:** Hyung Jun Park, Myeong Geun Choi.

**Writing – review & editing:** Hyung Jun Park, Jin-Young Huh, Ganghee Chae, Myeong Geun Choi.

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
