## [Decision Letter · Decision Letter 0]

13 Sep 2024

PONE-D-24-32590

Extraction of clinical data on major pulmonary diseases from unstructured radiologic reports using a large language model

PLOS ONE

Dear Dr. Choi,

Thank you for submitting your manuscript to PLOS ONE. After careful consideration, we feel that it has merit but does not fully meet PLOS ONE’s publication criteria as it currently stands. Therefore, we invite you to submit a revised version of the manuscript that addresses the points raised during the review process.

We look forward to receiving your revised manuscript.

Kind regards,

Harpreet Singh Grewal

Academic Editor

PLOS ONE

Journal Requirements:

2. We note that you have indicated that there are restrictions to data sharing for this study. PLOS only allows data to be available upon request if there are legal or ethical restrictions on sharing data publicly. For more information on unacceptable data access restrictions, please see http://journals.plos.org/plosone/s/data-availability#loc-unacceptable-data-access-restrictions. Before we proceed with your manuscript, please address the following prompts: a) If there are ethical or legal restrictions on sharing a de-identified data set, please explain them in detail (e.g., data contain potentially identifying or sensitive patient information, data are owned by a third-party organization, etc.) and who has imposed them (e.g., a Research Ethics Committee or Institutional Review Board, etc.). Please also provide contact information for a data access committee, ethics committee, or other institutional body to which data requests may be sent. b) If there are no restrictions, please upload the minimal anonymized data set necessary to replicate your study findings to a stable, public repository and provide us with the relevant URLs, DOIs, or accession numbers. For a list of recommended repositories, please see https://journals.plos.org/plosone/s/recommended-repositories. You also have the option of uploading the data as Supporting Information files, but we would recommend depositing data directly to a data repository if possible. We will update your Data Availability statement on your behalf to reflect the information you provide.

Additional Editor Comments:

<ol><li> 

**Handling of Medical Abbreviations and Acronyms**:

The manuscript should explicitly discuss whether and how the LLMs differentiated between medical abbreviations (e.g., "PNA" for pneumonia) and their full forms. This is crucial for understanding the model’s capability in interpreting radiologic reports where abbreviations are commonly used.<li> 

**Addressing Clinical Uncertainty**:

The manuscript must address how the LLMs handled clinical uncertainty present in radiologic reports, such as ambiguous diagnoses (e.g., "emphysema vs. COPD" or "lung cancer vs. metastasis"). A discussion on how the models interpreted these uncertainties and whether they required clinical correlation should be included.<li> 

**Interobserver Pulmonologist Qualifications**:

The qualifications of the pulmonologists involved in setting the gold standard need to be clearly stated. This includes their experience, training, and roles within their institutions. Additionally, the manuscript should discuss whether the LLMs accounted for differences between trainee and attending radiologist reads, which could impact the results.<li> 

**Error Rate of GPT-4**:

The reported error rate of 0% for GPT-4 in generating JSON formats seems implausible. The authors should re-examine this claim, providing a more nuanced explanation of what "error-free" means in this context and whether any minor or non-critical errors were overlooked.<li> 

**Introduction of JSON Format**:

The manuscript introduces the concept of JSON format without prior explanation, which may confuse readers. The authors should include a brief introduction to JSON formatting, particularly explaining its relevance and use in the study.<li> 

**Generalizability of Results**:

The study’s results are based on radiologic reports from Korean institutions, which may not generalize to other regions with different reporting styles. The manuscript should discuss this limitation and suggest future validation on more diverse datasets, including those from different languages and cultural contexts. The style of English usage in Korea may not be generalized elsewhere (especially in native English speaking countries). This remains a potential limitation of this study and should be discussed, especially when LLMs such as ChatGPT 4 seems to have heterogeneous performance in language translations in resource poor languages, as demonstrated in this recent study *https://www.mdpi.com/2075-4426/14/9/923*

<ol><li> 

**Challenges with ILD and Pneumonia Classification**:

The manuscript should expand on why the models, particularly GPT-4, had lower accuracy in diagnosing conditions like ILD and pneumonia. A discussion about the radiologic features of these diseases and the need for clinical correlation would help contextualize the models' performance.<li> 

**Clarification on Data Exclusion Criteria**:

The authors should provide a rationale for excluding reports with minimal findings, such as "no active lung lesions." Explaining how this exclusion might affect the model's learning and generalizability will strengthen the study's methodology section. This can also be added to the limitation section.<li> 

**Broader Dataset and Multimodal AI Research**:

Future research should consider incorporating datasets from multiple regions with diverse reporting styles. Additionally, exploring multimodal AI approaches that include image data alongside text could enhance the models' ability to classify complex conditions like ILD and pneumonia.<li> 

**Practical Implementation and Clinical Impact**:

A discussion on how these LLMs could be integrated into clinical workflows, particularly in reducing manual chart reviews without sacrificing diagnostic accuracy, would add practical value to the study

Reviewers' comments:

Reviewer's Responses to Questions

**Comments to the Author**

1. Is the manuscript technically sound, and do the data support the conclusions?

Reviewer #1: Yes

Reviewer #2: Yes

2. Has the statistical analysis been performed appropriately and rigorously? 

Reviewer #1: No

Reviewer #2: Yes

3. Have the authors made all data underlying the findings in their manuscript fully available?

Reviewer #1: Yes

Reviewer #2: Yes

4. Is the manuscript presented in an intelligible fashion and written in standard English?

Reviewer #1: Yes

Reviewer #2: Yes

5. Review Comments to the Author

Reviewer #1: The authors present an interesting manuscript to use a large language model to extract data on major pulmonary diseases from unstructured radiologic reports. The authors found that GPT-4 was the most accurate.

Regarding statistics, the authors need to explain in explicit detail how sensitivity, specificity, and accuracy were quantified in this study. This is not found in the statistical analysis section.

Is this large language model able to differentiate between acronyms vs. spelled out words (PNA for pneumonia, TB for tuberculosis, etc.)

How does this model account for clinical uncertainty? If a radiologist reads "emphysema vs. COPD" or "lung cancer vs. metastasis"...can the model account for these clinical obscurities as they often required clinical correlation.

What are the qualifications of the interobserver pulmonologists?

Does the language model take into account the read of a trainee vs. attending? Perhaps this model can help trainees better understand their shortcomings when over-reads by attendings have different reads or diagnoses.

The error rate of 0% from GPT-4 seems impossible as even attending pulmonologists are wrong a small percentage of the time when they read these images. There is also no introduction on JSON format before it is introduced in table 3 and makes it difficult to understand the meaning of that table.

The more applicable use of AI will be the interpretation of the images themselves vs. just the radiology reports.

Reviewer #2: This manuscript presents a compelling study on the application of large language models (LLMs), such as GPT-4, GPT-3.5, and Google's Gemini Pro, for extracting clinical data on major pulmonary diseases from unstructured radiologic reports. The study's design, methodology, and results highlight the potential for AI to enhance radiologic data interpretation, addressing a critical challenge in the healthcare field.

**Strengths:**

1. **Novel Application of LLMs**: The study explores a novel use case for LLMs, showing their capacity to accurately interpret radiological reports. This is a valuable addition to the growing literature on AI in healthcare, particularly in contexts where manual review is labor-intensive.

2. **High Accuracy and Specificity**: The results demonstrate that GPT-4, in particular, performed with impressive accuracy (0.89–0.99) and sensitivity across a range of pulmonary conditions, such as pleural effusion and emphysema. This confirms the utility of LLMs in clinical settings, especially for standardizing unstructured data.

3. **Ethical Considerations and Methodology**: The inclusion of ethical approval from multiple institutions and a clear explanation of the retrospective nature of the study, combined with the involvement of experienced pulmonologists, gives the study a robust foundation.

**Areas for Improvement:**

1. **Limited External Validation**: The study focuses on radiologic reports written by Korean radiologists, which may limit the generalizability of the findings to other regions where radiologic reporting styles differ significantly. Future research should consider validating these findings on more diverse datasets to assess whether the models perform consistently across languages and cultural differences in medical practice.

2. **Interpretation Challenges for Specific Conditions**: The models, particularly GPT-4, showed lower performance for diseases like interstitial lung disease (ILD) and pneumonia. It would be beneficial to provide a more in-depth discussion on why these specific conditions pose a challenge for LLMs, potentially incorporating additional medical context.

3. **Data Limitations**: The exclusion of cases with "no active lung lesions" or "no significant interval changes" may introduce a bias in the dataset. It would be helpful to include a rationale for this decision and how it may affect the study's overall results. Additionally, clarifying whether the inclusion of normal cases might help improve the models’ learning and prediction capabilities would be useful.

**Suggestions for Further Research:**

1. **Broader Dataset and Multimodal AI**: Incorporating datasets from multiple regions with diverse radiologic reporting styles and languages would provide a more comprehensive understanding of the LLMs' capabilities and limitations. Furthermore, exploring multimodal AI approaches, which incorporate image data along with text, could enhance the classification of complex conditions like ILD and pneumonia.

2. **Clinical Impact and Practical Implementation**: Future studies could explore the practical application of LLMs in clinical workflows, particularly how they can complement the work of radiologists. Assessing how these models can reduce time spent on manual chart reviews without sacrificing diagnostic accuracy will be critical for real-world implementation.

**Conclusion:**

The manuscript contributes significantly to the literature on AI in radiology, offering valuable insights into the application of LLMs for structured data extraction from unstructured reports. While there are some limitations regarding dataset diversity and the challenges posed by certain conditions, the study lays the groundwork for further advancements in AI-driven healthcare solutions.

** Minor Revision Suggestions:**

1. **Expand on the challenges with ILD and pneumonia classification**

**Section: Results (Page 8, Lines 169-170)**

Add more details explaining the reasons behind the lower accuracy in diagnosing interstitial lung disease (ILD) and pneumonia. It would be helpful to mention that both diseases often present with overlapping radiologic features that require clinical correlation for accurate diagnosis. You could also discuss how human expertise still plays a role in diagnosing these complex conditions and why AI models may struggle in such cases.

**Example addition:**

“The lower performance for conditions like ILD and pneumonia could be attributed to the complex and often overlapping radiological features of these diseases. For instance, ILD presents with varied findings such as interlobular septal thickening and fibrosis, which can be easily confused with other conditions like pulmonary edema or post-radiotherapy fibrosis. This emphasizes the need for clinical correlation, which AI models like GPT-4 currently lack, thus affecting their sensitivity and specificity in these cases.”

2. **Clarification on data exclusion criteria**

**Section: Methods (Page 7, Lines 129-132)**

Add a brief explanation as to why reports with "no active lung lesions" or other minimal findings were excluded. This will help address potential concerns about selection bias.

**Example addition:**

“These reports were excluded to focus the analysis on instances where pulmonary diseases were present, thus allowing the models to evaluate complex cases more effectively. While this exclusion helps concentrate the model's learning, it may also introduce selection bias, and future research should consider including such cases to improve the model’s generalization.”

3. **Include a note on dataset generalizability**

**Section: Discussion (Page 13, Lines 267-270)**

Briefly discuss the limitations of using a dataset that originates solely from Korean radiologists. Mention the need for future validation on a broader scale.

**Example addition:**

“While this study provides a solid foundation for evaluating LLMs in radiology, the dataset's origin—limited to Korean radiologists—may affect the generalizability of the findings. Different radiology practices and reporting styles across countries may influence the model’s performance. Therefore, validation on more diverse datasets, including reports written in various languages and medical contexts, is necessary to assess the broader applicability of the models.”

4. **Clarification on JSON format errors**

**Section: Results (Page 10, Lines 232-237)**

You mention error rates for JSON formatting but do not provide an explanation of the impact of these errors. A brief clarification would help contextualize this finding.

**Example addition:**

“While GPT-4 demonstrated a higher accuracy in producing error-free JSON format, the few errors observed in Gemini Pro 1.0 and GPT-3.5 primarily involved incorrect formatting or incomplete answers for specific diseases. These errors, although minor, highlight the importance of refining model outputs for seamless integration into clinical systems.”

6. PLOS authors have the option to publish the peer review history of their article (what does this mean?). If published, this will include your full peer review and any attached files.

Reviewer #1: No

Reviewer #2: No

---

## [Author Response · Author response to Decision Letter 0]

4 Nov 2024

October 27, 2024

Dear Editor:

We greatly appreciate the valuable comments provided for our manuscript entitled “Extraction of clinical data on major pulmonary diseases from unstructured radiologic reports using a large language model.”

We are certain that the valuable suggestions from the reviewers have helped improve the quality of the manuscript. We have revised the manuscript according to the reviewers’ recommendations. The revised text has been indicated in blue font. Our point-by-point responses to all comments have been included below.

We hope that the revised manuscript is now suitable for publication in PLOS One as a Research Article.

Sincerely,

Myeong Geun Choi, MD, PhD

Division of Pulmonary and Critical Care Medicine, Mokdong hospital, College of Medicine, Ewha Womans University, 1071 Anyangcheon-Ro, Yangcheon-gu, Seoul, 07985, Korea 

Tel: +82-2-2650-5417

Fax: +82-2-2650-5272

E-mail: cmkcmk1006@gmail.com

Ganghee Chae, MD

Division of Respiratory and Critical Care Medicine, Department of Internal Medicine, Ulsan University Hospital, University of Ulsan College of Medicine, 25 Daehakbyeongwon-ro, Dong-gu, Ulsan, 44033, South Korea

Tel: +82-52-250-7029

Fax: +82-52-250-7048

E-mail: ganghee@uuh.ulsan.kr

Editor: 

Hard Recommendations (Must Address Before Acceptance):

Q1. Handling of Medical Abbreviations and Acronyms:

The manuscript should explicitly discuss whether and how the LLMs differentiated between medical abbreviations (e.g., “PNA” for pneumonia) and their full forms. This is crucial for understanding the model’s capability in interpreting radiologic reports where abbreviations are commonly used. 

Reply: Thank you for raising this important question about the language models’ ability to handle abbreviations and spelled-out words. We would like to clarify that the LLMs used in our study (Gemini Pro 1.0, GPT-3.5, and GPT-4) are indeed capable of understanding and differentiating between abbreviations and their full forms in context.

These models are trained on vast amounts of text data, including medical literature, which enables them to recognize common medical abbreviations like PNA for pneumonia and TB for tuberculosis, as well as their spelled-out versions. The models interpret these terms within the context of the entire radiologic report, allowing accurate inferences to be made regardless of whether abbreviations or full terms are used. We did not perform any specific preprocessing to standardize acronym usage, as the models demonstrated the ability to naturally handle this variation. This capability is part of the models’ strength in processing natural language as it appears in real-world medical reports, contributing to their effectiveness in our study.

Changes in the text: Lines 303–310

“Furthermore, the LLM interpreted medical abbreviations within the context of the entire radiologic report, making inferences regardless of whether abbreviations or full terms were used. For instance, GPT-4 correctly recognized all instances of “TB” as “tuberculosis” in the reports. We did not perform any specific preprocessing to standardize acronym usage, as the models demonstrated the ability to handle this variation naturally. These capabilities are part of the models’ strength in processing natural language as it appears in real-world medical reports, contributing to its effectiveness in our study.”

Q2. Addressing Clinical Uncertainty:

The manuscript must address how the LLMs handled clinical uncertainty present in radiologic reports, such as ambiguous diagnoses (e.g., “emphysema vs. COPD” or “lung cancer vs. metastasis”). A discussion on how the models interpreted these uncertainties and whether they required clinical correlation should be included.

Reply: We appreciate this valuable comment regarding how LLMs handle clinical uncertainty in radiologic reports. The models we used were pre-trained on various medical literature and seem to avoid making judgments on items with clinical uncertainty, that is, items in the radiologic report lacking definitive evidence for diagnosis. For example, upon reviewing the decisions made by GPT-4, it did not classify “developing and increasing in size of heterogeneously enhancing nodules in the right lower lobe“ as lung cancer on a CT scan of a patient with breast cancer. In a case where the report mentioned the possibility of chronic obstructive pulmonary disease, a condition where emphysema may appear in radiologic findings but is not essential for diagnosis, in a patient with chronic bronchitis, GPT-4 did not classify the finding as emphysema.

Changes in the text: Lines 295–303

“On the other hand, in some cases, it appeared that the LLM correlated diagnostic criteria with the context of the radiologic report and avoided making judgments on items with clinical uncertainty, that is, items in the report lacking definitive evidence for diagnosis. For example, upon reviewing the decisions made by GPT-4, it did not classify “developing and increasing in size of heterogeneously enhancing nodules in the right lower lobe” as lung cancer on a CT scan of a patient with breast cancer. In a case where the report mentioned the possibility of chronic obstructive pulmonary disease, a condition where emphysema may appear in radiologic findings but is not essential for diagnosis [24], in a patient with chronic bronchitis, GPT-4 did not classify the finding as emphysema.”

Q3. Interobserver Pulmonologist Qualifications:

The qualifications of the pulmonologists involved in setting the gold standard need to be clearly stated. This includes their experience, training, and roles within their institutions. Additionally, the manuscript should discuss whether the LLMs accounted for differences between trainee and attending radiologist reads, which could impact the results.

Reply: Thank you for your valuable comment. The three pulmonologists who performed the labeling have over 10 years each of experience as physicians, including more than 5 years each of clinical experience as pulmonologists. All three hold assistant professorships at university hospitals. GC’s clinical and research areas are tuberculosis (mycobacterial disease), lung cancer, and interventional pulmonology. MGC specializes in lung cancer and interventional pulmonology, and J-YH’s areas of expertise are lung cancer and interstitial lung disease. These details have been added to the Methods section.

Changes in the text: Lines 118–131

“The following outcomes were defined: pneumonia, interstitial lung disease (ILD), active pulmonary tuberculosis (TB), pulmonary edema, pleural effusion, lung cancer, and emphysema. On CT scans, interstitial lung abnormalities were identified as ILD, and lung abscesses were classified as pneumonia. The complete resolution and improvement were recorded as absent, whereas interval improvement and decreases were recorded as present. On chest radiographs, pulmonary congestion was assessed for edema, and interstitial pneumonia was categorized as ILD. Three pulmonology specialists, GC, MGC, and J-YH, each with over a decade of experience as practicing physicians including more than 5 years each of clinical experience in pulmonology independently reviewed the reports and labeled the presence or absence of seven diseases. All three hold assistant professorships at university hospitals with clinical and research expertise. GC’s clinical and research areas include tuberculosis (mycobacterial disease), lung cancer, and interventional pulmonology; MGC specializes in lung cancer and interventional pulmonology; and J-YH’s areas of expertise are lung cancer and interstitial lung disease. The gold standard was established as an agreement among at least two labelers.”

Q4. Error Rate of GPT-4:

The reported error rate of 0% for GPT-4 in generating JSON formats seems implausible. The authors should re-examine this claim, providing a more nuanced explanation of what “error-free” means in this context and whether any minor or non-critical errors were overlooked.

Reply: We appreciate the astute observation regarding the reported 0% error rate for GPT-4 in generating JSON formats. We acknowledge that this result warrants further explanation and have added clarification to our Methods and Discussion sections.

In our study, we defined “error-free“ JSON formats as outputs that strictly adhered to the specified structure, including all seven disease fields with appropriate “yes” or “no” values, without any syntactical errors or missing information. We have added this definition to our Discussion section for clarity. While GPT-4 achieved perfect structural accuracy in our dataset, we acknowledge that this does not reflect potential semantic errors in disease classification, which are separately addressed in our performance analysis.

Changes in the text: Lines 325–330

“Although all AI models demonstrated strong agreement with human annotators in terms of performance, notable discrepancies were observed in their stability when generating JSON formats. GPT-4 showcased exceptional stability, flawlessly producing error-free JSON formats on its first attempt. Error-free JSON formats refer to outputs that strictly adhered to the specified structure, including all seven disease fields with appropriate “yes” or “no” values, without any syntactical errors or missing information.”

Q5. Introduction of JSON Format:

The manuscript introduces the concept of JSON format without prior explanation, which may confuse readers. The authors should include a brief introduction to JSON formatting, particularly explaining its relevance and use in the study.

Reply: We thank the reviewer for pointing out the need for a clearer introduction to JSON formatting in our manuscript. We agree that this is an important aspect of our methodology that requires explanation for readers who may be unfamiliar with the concept.

To address this, we have added a brief explanation of JSON formatting and its relevance to our study in the Statistical Analysis section of the Methods. This addition provides context for our use of JSON and explains its role in evaluating the models’ performance.

Changes in the text: Lines 150–156

“Additionally, the accuracy of each label was evaluated against the gold standard using these metrics, stratified by disease and hospital, to determine the accuracy and appropriateness of human annotation. Interobserver agreement among the labelers was assessed by calculating the Fleiss kappa value for each disease. The output of each model was evaluated for adherence to the specified JSON format, which included the correct structure and presence of all seven disease fields. Errors in JSON formatting were recorded and analyzed to assess the models’ ability to consistently produce structured output.”

Q6. Generalizability of Results:

The study’s results are based on radiologic reports from Korean institutions, which may not generalize to other regions with different reporting styles. The manuscript should discuss this limitation and suggest future validation on more diverse datasets, including those from different languages and cultural contexts. The style of English usage in Korea may not be generalized elsewhere (especially in native English speaking countries). This remains a potential limitation of this study and should be discussed, especially when LLMs such as ChatGPT 4 seems to have heterogeneous performance in language translations in resource poor languages, as demonstrated in this recent study https://www.mdpi.com/2075-4426/14/9/923

Reply: Thank you for pointing out the potential limitation regarding external validation of our study. We acknowledge that our study utilized radiologic reports written by Korean radiologists, which may limit its generalizability to other regions where reporting styles and medical practices may differ. As suggested, future studies could benefit from validating the findings using datasets from different countries and linguistic backgrounds to determine the models’ performance across various contexts. We have added this point to the discussion of our study’s limitations.

Changes in the text: Lines 346–356

“Moreover, this research utilized radiologic reports written by Korean radiologists, which may limit its generalizability to regions with different reporting styles and medical practices [28]. Future studies could benefit from validating these findings using datasets from other countries and linguistic backgrounds to assess the models’ performance across diverse contexts. Lastly, the models’ performance in this study may have been slightly compromised due to the lack of recent advancements in prompt design techniques. The potential impact of prompt design approaches such as chain of thought, reflection, and few-shot in-context learning, which have shown promise in improving model performance, was not evaluated in this study. Consequently, further research is required to validate these prompt design techniques and enhance the accuracy and reliability of AI-assisted classification in medical contexts.”

Suggestions (Improvements to Strengthen the Manuscript):

Q7. Challenges with ILD and Pneumonia Classification:

The manuscript should expand on why the models, particularly GPT-4, had lower accuracy in diagnosing conditions like ILD and pneumonia. A discussion about the radiologic features of these diseases and the need for clinical correlation would help contextualize the models’ performance.

Reply: Thank you for your insightful feedback. Diagnosing ILD and pneumonia based solely on imaging findings is challenging. ILD can be classified into many different types depending on the cause, clinical presentation, and histological findings. The imaging spectrum is highly variable, including honeycombing, reticular opacity, traction bronchiectasis, fibrosis, ground-glass opacity, consolidation, nodules, and cystic lesions (Ref. 1). However, similar imaging findings, such as fibrosis, ground-glass opacity, and consolidation, can also appear in cases of infection, drug reactions, or radiation therapy. Thus, a comprehensive assessment of the clinical context is essential when confirming ILD. Similarly, pneumonia cannot be diagnosed based solely on imaging findings and must be accompanied by clinical signs and symptoms (Ref. 2). This highlights that LLMs are not yet fully capable of considering clinical correlations, which may explain the reduced accuracy of the LLM models in these conditions.

We analyzed cases categorized by GPT-4 as ILD but diagnosed otherwise by pulmonologists . The most common cases, accounting for 53.7% of these instances, occurred when GPT-4 classified any findings of fibrosis on chest CT, regardless of the cause (e.g., radiotherapy, COVID, tuberculosis), as ILD, whereas pulmonologists did not. Additionally, due to the challenge of diagnosing ILD solely on one finding, pulmonologists did not classify cases of interlobular septal thickening, which could also indicate pulmonary edema or lymphangitic metastasis, as ILD. Conversely, GPT-4 classified them as ILD (22.1%).

We have addressed this in detail in the Discussion section.

References

1. Raghu G, Remy-Jardin M, Richeldi L, Thomson CC, Inoue Y, Johkoh T, et al. Idiopathic Pulmonary Fibrosis (an Update) and Progressive Pulmonary Fibrosis in Adults: An Official ATS/ERS/JRS/ALAT Clinical Practice Guideline. Am J Respir Crit Care Med. 2022; 205: e18-e47.

2. Metlay JP, Waterer GW, Long AC, Anzueto A, Brozek J, Crothers K, et al. Diagnosis and Treatment of Adults with Community-acquired Pneumonia. An Official Clinical Practice Guideline of the American Thoracic Society and Infectious Diseases Society of America. Am J Respir Crit Care Med. 2019; 200: e45-e67.

Changes in the text: Lines 267–277

“For ILD, the specificity was relatively low (0.89 in X-ray and 0.88 in chest CT). ILD can be classified into many different types depending on the cause, clinical presentation, and histological findings. The imaging spectrum is highly variable, including honeycombing, reticular opacity, traction bronchiectasis, fibrosis, ground-glass opacity, consolidation, nodules, and cystic lesions [20]. However, similar imaging findings, such as fibrosis, ground-glass opacity, and consolidation, can also appear in cases of infection, drug reactions, or radiation thera

---

## [Editor Report · Decision Letter 1]

6 Nov 2024

Extraction of clinical data on major pulmonary diseases from unstructured radiologic reports using a large language model

PONE-D-24-32590R1

Dear Dr. Choi,

We’re pleased to inform you that your manuscript has been judged scientifically suitable for publication and will be formally accepted for publication once it meets all outstanding technical requirements.

Kind regards,

Harpreet Singh Grewal

Academic Editor

PLOS ONE

Additional Editor Comments (optional):

please expand JSON at its first mention. this has not been addressed yet.
---

## [Editor Report · Acceptance letter]

13 Nov 2024

PONE-D-24-32590R1 

PLOS ONE

Dear Dr. Choi, 

I'm pleased to inform you that your manuscript has been deemed suitable for publication in PLOS ONE. Congratulations! Your manuscript is now being handed over to our production team.

Kind regards, 

on behalf of

Dr. Harpreet Singh Grewal 

Academic Editor

PLOS ONE